# Developing a Supplementary Diagnostic Tool for Breast Cancer Risk Estimation Using Ensemble Transfer Learning

**DOI:** 10.3390/diagnostics13101780

**Published:** 2023-05-18

**Authors:** Tengku Muhammad Hanis, Nur Intan Raihana Ruhaiyem, Wan Nor Arifin, Juhara Haron, Wan Faiziah Wan Abdul Rahman, Rosni Abdullah, Kamarul Imran Musa

**Affiliations:** 1Department of Community Medicine, School of Medical Sciences, Universiti Sains Malaysia, Kubang Kerian 16150, Kelantan, Malaysia; 2School of Computer Sciences, Universiti Sains Malaysia, Gelugor 11800, Penang, Malaysia; intanraihana@usm.my (N.I.R.R.); rosni@usm.my (R.A.); 3Biostatistics and Research Methodology Unit, School of Medical Sciences, Universiti Sains Malaysia, Kubang Kerian 16150, Kelantan, Malaysia; wnarifin@usm.my; 4Department of Radiology, School of Medical Sciences, Universiti Sains Malaysia, Kubang Kerian 16150, Kelantan, Malaysia; drjuhara@usm.my; 5Breast Cancer Awareness and Research Unit, Hospital Universiti Sains Malaysia, Kubang Kerian 16150, Kelantan, Malaysia; wfaiziah@usm.my; 6Department of Pathology, School of Medical Sciences, Universiti Sains Malaysia, Kubang Kerian 16150, Kelantan, Malaysia

**Keywords:** Asian women, breast cancer, transfer learning, deep learning, artificial intelligence, diagnostic screening, mammography, radiologists

## Abstract

Breast cancer is the most prevalent cancer worldwide. Thus, it is necessary to improve the efficiency of the medical workflow of the disease. Therefore, this study aims to develop a supplementary diagnostic tool for radiologists using ensemble transfer learning and digital mammograms. The digital mammograms and their associated information were collected from the department of radiology and pathology at Hospital Universiti Sains Malaysia. Thirteen pre-trained networks were selected and tested in this study. ResNet101V2 and ResNet152 had the highest mean PR-AUC, MobileNetV3Small and ResNet152 had the highest mean precision, ResNet101 had the highest mean F1 score, and ResNet152 and ResNet152V2 had the highest mean Youden J index. Subsequently, three ensemble models were developed using the top three pre-trained networks whose ranking was based on PR-AUC values, precision, and F1 scores. The final ensemble model, which consisted of Resnet101, Resnet152, and ResNet50V2, had a mean precision value, F1 score, and Youden J index of 0.82, 0.68, and 0.12, respectively. Additionally, the final model demonstrated balanced performance across mammographic density. In conclusion, this study demonstrates the good performance of ensemble transfer learning and digital mammograms in breast cancer risk estimation. This model can be utilised as a supplementary diagnostic tool for radiologists, thus reducing their workloads and further improving the medical workflow in the screening and diagnosis of breast cancer.

## 1. Introduction

Breast cancer is the most commonly diagnosed cancer worldwide [1]. Breast cancer is considered the leading cause of cancer-related death in the twelve regions of the world [2]. This disease accounts for one in four and one in six cancer cases and cancer deaths among women, respectively [3]. In an attempt to combat the disease, the World Health Organization (WHO) proposed a global breast cancer initiative in 2021, which will run over 20 years and consist of three key elements [4]. One of these three key elements is the promotion of the early detection of breast cancer. The early detection of this disease ensures that a patient receives timely treatment. Thus, any delay in the medical workflow of breast cancer screening and diagnosis will influence the prognosis of the disease.

Artificial intelligence (AI) is expected to improve the efficiency of the healthcare system, including in the areas of oncology and radiology. Researchers have studied the use of AI in thoracic imaging, abdominal and pelvic imaging, colonoscopy, mammography, brain imaging, and radiation oncology [5]. Digital mammograms have been widely used as part of breast cancer assessment. The use of screening mammograms has been shown to improve the early detection of breast cancer, which, in turn, reduces breast cancer mortality [6]. The introduction of mammogram-related AI to assist radiologists in breast cancer assessment may reduce their workload and further improve the diagnostic accuracy of mammogram readings. Additionally, such programs will provide radiologists with greater availability to engage and focus on more complex medical cases or higher-level tasks. In fact, AI has been shown to reduce the time required by radiologists to interpret mammograms, thereby improving overall cancer detection [7].

Transfer learning, or a pre-trained network, constitutes a network previously trained on a large dataset [8]. The use of pre-trained networks is expected to reduce the training time and improve the overall performance of deep learning tasks [9]. The early layer of the convolutional neural network (CNN) learn to recognize the general and broader aspects of an image, such as edges, textures, and patterns, while the last few layers learn to recognize the more specific features of the image related to the task [10]. Hence, the main idea of transfer learning is to transfer the layers learned early on, and trained through one task, to another. There are two approaches to implementing transfer learning: (1) feature extraction and (2) fine tuning. The former allows the previously trained network to be used on a different task without the need to train from scratch, while the latter allows for some adjustments to the pre-trained network by unfreezing a few final layers. The fine-tuned approach allows the pre-trained network to adapt to the new task and may further improve its performance in the task.

Transfer learning has been applied to medical image analysis of areas such as the brain, lungs, kidneys, skin, colon, and breasts [11]. Several pre-trained networks are commonly used, including VGGNet and its variants, ResNet and its variants, MobileNet and its variants, and NASNet and its variants. VGGNet was proposed by the Visual Geometry Group (VGG) in 2015 [12]. VGGNet consists of two variants: VGG16 and VGG19. Both models are an improvement from AlexNet and use several small kernel-sized filters instead of large kernel-sized filters. Thus, the proposed VGG16 and VGG19 networks contained 13 and 16 convolutional layers, respectively. Additionally, in 2015, ResNet, which incorporates residual learning, was introduced [13]. ResNet overcame the issue of vanishing and exploding gradients due to an increased number of network layers. MobileNet was introduced in 2017 by Google researchers [14]. The network was designed to be smaller and less computationally expensive but without sacrificing performance. In 2018, another team of Google researchers, Google Brain, presented a NASNet architecture [15]. The architecture utilised a NASNet search space, which was a new search space design coupled with a new regularisation technique known as *ScheduledDropPath*. Consequently, NASNet was able to achieve excellent performance with smaller network layers and lower complexity.

Ensemble transfer learning combines several transfer-learning candidates to achieve better performance. Ensembling involves aggregating the individual predictions of the candidate models to achieve a more accurate and robust prediction [8]. Furthermore, ensemble learning has been suggested to be one of the approaches capable of mitigating the class imbalance issue [16]. In recent years, the ensemble learning model has presented good performance in the field of medicine and healthcare [17]. The application of ensemble transfer learning has been studied with respect to its use for the detection of dental caries [18], the detection of COVID-19 [19], the classification of skin lesions [20], the classification of histopathology images [20], the diagnosis and prognosis of Alzheimer’s disease [21], and the determination of drug response in major depressive disorder [22].

When developing a robust model for breast cancer classification, factors influencing the performance of the model should be considered. One of the important risk factors of breast cancer is mammographic density [23]. Mammographic density or breast density indicates the amount of dense tissue in a breast. Mammographic density influences the risk of breast cancer and affects the sensitivity of mammograms [24,25]. The objective of this study is to develop a supplementary diagnostic tool for radiologists. Therefore, this study will explore the use of ensemble pre-trained networks and digital mammograms for breast cancer risk estimation. The performance of the model will be further evaluated across a range of mammographic densities.

## 2. Related Works

Several studies have been conducted related to the application of transfer learning to digital mammograms for breast cancer classification. Saber et al. [26] explored the use of six pre-trained networks for breast cancer classification. The study managed to achieve an accuracy of 0.99, wherein VGG16 was identified as the best-performing model. Another study published in the same year explored the use of a hybrid model by combining a modified VGG16 network and ImageNet, which managed to achieve an accuracy of 0.94 [27]. Several other studies managed to achieve good performance with both VGG16 and VGG19 [28,29,30]. In addition, a study by Guan and Loew [31] comparing the feature extraction and fine-tuning approaches using VGG16 showed that the latter performed better compared to the former; however, the difference in performance was very minimal.

Several studies have explored the use of ResNet for breast cancer detection. Yu and Wang [32] compared several ResNet models, including ResNet18, ResNet50, and ResNet101, in their study. Consequently, it was determined that ResNet18 had the highest accuracy at 0.96, outperforming all the other ResNet variants. Another study compared several pre-trained networks, including ResNet50, NASNet, InceptionV3, and MobileNet [33]. Essentially, this study applied two different pre-processing approaches to the mammogram images. Otsu thresholding was not applied in the first approach but was applied in the second approach. ResNet50 was the best model in the first approach with an accuracy of 0.78, while NASNet was the best model in the second approach with an accuracy of 0.68.

Additionally, a study by Ansar [34] proposed a transfer learning network using a MobileNet architecture for breast cancer classification. This study utilised two datasets separately, namely, the Digital Database for Screening Mammography (DDSM) and the curated breast imaging subset of DDSM (CBIS-DDSM), and achieved accuracies of 0.87 and 0.75, respectively. Therefore, the result of this study suggests the use of different datasets may influence the performance of a transfer learning model.

Furthermore, other pre-trained network architectures have been analysed with respect to their performance in breast cancer classification using digital mammograms. Jiang et al. [35] compared transfer learning models and deep learning models trained from scratch and compared the performance of GoogleNet and AlexNet in terms of breast cancer classification. The study reported that transfer learning and GoogleNet outperformed the other network. Another study explored the application of the InceptionV3 architecture to the INBreast dataset, for which the highest AUC was achieved at 0.91 [36]. Recently, a study by Pattanaik et al. [37] proposed a hybrid transfer learning model consisting of DenseNet121 and an extreme learning machine (ELM). The model achieved an accuracy of 0.97 and outperformed the other models in the study. Table 1 presents the summary of previous works related to pre-trained networks and breast cancer classification that utilised digital mammograms. Notably, aside from that conducted by Mendel et al., all the aforementioned studies utilised publicly available datasets [29].

## 3. Materials and Methods

### 3.1. Data

Two datasets were utilised in this study. Digital mammograms and their reports were retrieved from the department of radiology, Hospital Universiti Sains Malaysia (HUSM), and histopathological examination (HPE) results were retrieved from the department of pathology, HUSM. Generally, each set of mammogram images may consist of the right and left sides of a breast. Each side may consist of mediolateral oblique and craniocaudal views. Additionally, the mammogram reports contained information on the Breast-Imaging-Reporting and Data System (BI-RADS) breast densities and classifications, while the HPE results contained information on the classification of the breast lesions. The data were collected from 1 January 2014 until 30 June 2020 from each respective department. Next, the two datasets were combined if the HPE data dated from within a year after the mammogram was taken.

BI-RADS breast density information was used to split the mammograms into non-dense and dense breasts. The non-dense breast cases consisted of BI-RADS densities of A and B, while the dense breast cases consisted of BI-RADS densities of C and D. Each mammogram was classified as either normal or suspicious and labelled accordingly. A normal mammogram was a mammogram with a BI-RADS classification of 1 or that was reported normal according to the HPE result. A suspicious mammogram was a mammogram with BI-RADS classification of 2, 3, 4, 5, or 6, or one that was reported as benign or malignant according to the HPE result. Additionally, a mammogram with a BI-RADS classification of 0 was excluded from this study. Overall, there were 7452 mammograms utilised in this study. About 1651 mammograms corresponded to the normal class, while 5801 mammograms corresponded to the suspicious class. Figure 1 presents a sample of normal and suspicious mammograms in non-dense and dense groups. Breast density was used in the model evaluation process and not in the model development process to ensure the generalisability of the model.

### 3.2. Pre-Processing Steps

Each mammogram was pre-processed using a median filter, Otsu thresholding [38], and contrast-limited adapted histogram equalisation (CLAHE). A median filter is a non-linear filtering method that is used to remove noise in an image. Concerning mammograms, several studies have shown that median filters present good performance with respect to preserving the sharp edges of images and that they are robust to outliers [39,40,41]. Otsu thresholding is a type of clustering-based image-thresholding technique used to binarize an image based on pixel intensities. This method has been shown to successfully remove unwanted regions of high intensities and the pectoral muscle in mammograms, thus further improving mammogram classification and breast cancer detection [42,43]. Additionally, CLAHE was utilised to enhance the contrast of the mammogram. Several studies have proposed the use of this method as a pre-processing technique to improve the predictive performance of breast cancer detection [44,45,46]. Lastly, the mammograms were rescaled, resized to 480 × 480, and their format was changed from DICOM to JPEG to reduce the size of the mammograms. Figure 2 illustrates the general flow of the image pre-processing procedure applied to the mammograms.

All the pre-processing steps were performed in R version 4.2.1 [47]. The *reticulate* [48] and *pydicom* [49] packages were used to read the mammogram into R. The *nandb* [50], *EBImage* [51], and *autothresholdr* [52] packages were used to implement the median filter, perform CLAHE and resizing of the mammograms, and apply Otsu thresholding to the mammograms, respectively.

### 3.3. Pre-Trained Network Architecture

Thirteen pre-trained network architectures were selected based on previous studies (Table 1), including MobileNets [14], MobileNetV2 [53], MobileNetV3Small [14], NASNetLarge [15], NASNetMobile [15], ResNet101 [13], ResNet101V2 [54], ResNet152 [13], ResNet152V2 [54], ResNet50 [13], ResNet50V2 [54], VGG16 [12], and VGG19 [12]. All pre-trained networks were run in R using *keras* [55] and *tensorflow* [56] packages. The pre-trained networks were designed to classify the mammogram images into normal and suspicious classes.

The fine-tuning approach was used to customise the pre-trained network. The top layer with the largest parameters was unfrozen layer by layer. The process would stop once a pre-trained network with a currently unfrozen layer could not achieve better performance than a pre-trained network with an unfrozen previous layer.

### 3.4. Model Development and Comparison

The data were split into three training–testing splits: (1) 70–30%, (2) 80–20%, and (3) 90–10%. The validation dataset was set to 10% of each training dataset. Each mammogram was randomly classified into training, validation, and testing datasets. However, two stratification factors were taken into consideration: the distribution of the breast density and mammogram classification. Thus, each training dataset, validation dataset, and testing dataset in each split was equally stratified and had an equal proportion of breast densities (dense and non-dense) and mammogram classifications (normal and suspicious).

Data augmentation and dropout were applied to overcome overfitting. Each mammogram was randomly flipped along its horizontal axis, rotated by a factor of 0.2 radians, and zoomed in or out by a factor of 0.05. The dropout rate was set to 0.5. Additionally, class weight was used to overcome the class imbalance between normal and suspicious cases. The ratio of class weights used was 2.26 for normal and 0.64 for suspicious cases. Thus, the loss function heavily penalised the misclassification of the minority class (normal cases) compared to the misclassification of the majority class (suspicious cases). Binary cross-entropy was used as a loss function, and the Adam [57] algorithm was used as an optimiser. The learning rate was set to 1 × 10^−5^. Lastly, a sigmoid activation function was used in the last layer to determine the probability of the mammogram being suspicious. The network with the highest precision–recall area under the curve (PR-AUC) on the validation dataset was selected as the final model for each pre-trained network.

The evaluation criteria were applied to determine the top fine-tuned, pre-trained networks. The evaluation criteria utilised were a Youden J index > 0 and F1 score > 0.6. The candidates for the ensemble model were selected based on the PR-AUC, precision, and F1 score. Each ensemble model consisted of the top three pre-trained networks based on the three aforementioned performance metrics. The majority voting approach was utilised in each ensemble model to determine the final prediction.

### 3.5. Performance Metrics

Generally, the six performance metrics used in this study were PR-AUC, precision, F1 score, Youden J index, sensitivity, and specificity. The accuracy and the receiver operating characteristic area under the curve (ROC-AUC) were not used in this study due to the imbalanced nature of the dataset. The two metrics were not appropriate and less informative for the imbalanced dataset [58,59]. The performance metrics utilised in this study are defined below:$$Precision=\frac{TP}{TP+FP}$$
$$F1\;\text{score}=2\times \frac{\text{precision}\times \text{recall}}{\text{precision}+\text{recall}}$$
Youden J index=sensitivity+specificity−1
$$\text{Recall}/\text{sensitivity}=\frac{TP}{TP+FN}$$
$$\text{Specificity}=\frac{TN}{TN+FP}$$

A true positive case was defined as a suspicious case that was predicted to be suspicious by the network, while a true negative case was a normal case that was predicted to be normal by the network. A false negative case was a suspicious case that was predicted to be normal by the network, while a false positive case was a normal case that was predicted to be suspicious by the network. All six performance metrics were aggregated across the three different splits and presented as mean and standard deviation (SD).

### 3.6. Performance across Breast Densities

The final ensemble model was evaluated using the overall, dense, and non-dense testing datasets. The performance metrics were compared statistically using the Wilcoxon rank sum statistical test. A *p* value < 0.05 indicated that there was a significant difference in performance metrics between the dense and non-dense cases. Figure 3 illustrates the overall flow of the analysis in this study.

## 4. Results

### 4.1. Model Development

In this study, thirteen pre-trained networks were developed and fine-tuned for breast abnormality detection. The pre-trained networks were selected based on previous studies (Table 1). Table 2 presents all the network architectures utilised in this study. The networks with the highest means in terms of PR-AUC, precision, F1 score, and the Youden J index were ResNet101V2 and ResNet152, MobileNetV3Small and ResNet152, ResNet101, and ResNet152 and ResNet152V2, respectively. After the application of the evaluation criteria (refer to Section 3.4), only six networks remained out of the thirteen pre-trained networks. Figure 4 presents all six selected pre-trained networks.

### 4.2. Ensemble Transfer Learning

Three ensemble models were developed using a majority-voting approach. Ensemble model 1 consisted of Resnet101, NASNetMobile, and ResNet50V2. Ensemble model 2 consisted of Resnet101V2, Resnet152, and ResNet50V2. Finally, ensemble model 3 consisted of Resnet101, Resnet152, and ResNet50V2. Ensemble models 1, 2, and 3 were developed based on the top F1 scores, PR-AUC values, and precision scores, respectively. Table 3 compares the performance metrics of the ensemble models and each candidate network. Ensemble model 3 had the highest mean precision and Youden J index, while ResNet101 had the highest mean F1 score. Thus, ensemble model 3 was selected as the final model in this study.

### 4.3. Performance across Breast Densities

The final ensemble model consisted of Resnet101, Resnet152, and ResNet50V2. The performance of the final ensemble model was evaluated using three datasets: overall, dense, and non-dense testing datasets. Table 4 presents the descriptive performance of the model across the three testing datasets, while Table 5 presents the result of the performance comparison of the model across dense and non-dense breast cases using the Wilcoxon rank sum statistical test. The final model had slightly higher performance metrics in the dense breast cases compared to the non-dense breast cases (Table 4). However, all the *p* values in Table 5 are above 0.05. Thus, the result of the Wilcoxon rank sum statistical test indicated that there was no significant difference between the dense and non-dense breasts across all performance metrics.

## 5. Discussion

The final ensemble model in this study displayed good performance with a precision value of 0.82. Several studies have achieved better precision metrics compared to those presented in this study, ranging from 0.84 to 0.97 [26,27,34]. However, all these studies utilised publicly available datasets. Studies that use publicly available datasets have been shown to have better performance compared to those that use primary datasets [60]. The data utilised in this study were mildly imbalanced. The proportion of minority class or normal mammograms amounted to 22% of the total dataset. Thus, commonly used performance metrics such as accuracy and ROC-AUC were not appropriate in this study [58,59]. However, the data used in this study were collected from a hospital’s department of radiology and pathology. Therefore, the performance presented in this study is more realistic and reflective of the actual performance of the deep learning model with respect to mammographic data for breast abnormality detection. Notably, the performance of the final ensemble model was just slightly better than the initial fine-tuned pre-trained networks, especially compared to MobileNetV3Small and ResNet152 (results in Table 2 and Table 3). However, a study by Khan et al. [61] wherein an ensemble pre-trained network was implemented showed better performance. The study utilised a microscopic image dataset to classify breast cancer and reported an accuracy of 0.98 for their ensemble transfer learning model. The average accuracy of candidate transfer learning model was 0.94. On the other hand, a study by Zheng et al. [62] that applied ensemble transfer learning to classify breast cancer displayed minimal performance improvement. The study utilised microscopic biopsy images and achieved an accuracy of 0.989 for its ensemble model. The highest accuracy of the candidate model in the study was 0.988.

This final ensemble model in this study also presented balanced performance between specificity and sensitivity with an F1 score of 0.68. Theoretically, the relationship between the two early metrics is inversely proportionate [63]. A diagnostic tool with high sensitivity typically has low specificity, and vice versa. Thus, balanced performance between the metrics was preferred; however, any cut-off values have yet to be established. A further evaluation of the ensemble model across breast densities revealed that there was no significant performance difference between dense and non-dense cases (Table 5). In previous studies, it was shown that particularly high mammographic densities reduced the sensitivity of mammograms and increased the risk of breast cancer [64,65]. Since Asian women tend to have denser breasts compared to other ethnicities [66], this factor plays a significant role in the screening and diagnosis of breast cancer in this population. The performance of any screening or diagnostic tool that utilises mammography should be evaluated with respect to breast density.

Digital mammograms have been widely used in the initial screening of breast cancer [67]. Thus, this study utilised digital mammograms to develop an ensemble transfer learning model that can be used by radiologists as a supplementary diagnostic tool. Other types of data that have been used to predict or classify breast cancer include imaging modalities and tabular data. Tabular data include medical records, socio-demographic information, and clinical information, whereas imaging modalities include mammograms, digital breast tomosynthesis (DBT), ultrasound images, computed tomography, positron emission tomography, magnetic resonance imaging (MRI), and thermography. The selection of the appropriate type of data for the development of a machine learning model depends on the objective of the study and the stage at which the model will be utilised in the breast cancer medical workflow. A deep-learning-based prognostic model may utilise more advanced and confirmative imaging modalities such as DBT or histopathological images. However, the use of more advanced imaging modalities such as DBT and MRI may limit the applicability of the developed deep learning model to larger medical facilities or research centres where such equipment is exclusively available.

This study utilised mammographic data collected from a university-based hospital. The data were further evaluated by a radiologist and a pathologist. Thus, the data utilised in this study were of high quality and reflective of the actual cases in the hospital. Despite these strengths, this study suffered mild imbalanced classification. Hence, common performance metrics such as accuracy and ROC-AUC were not appropriate for use in this study. Consequently, the utilisation of different performance metrics rendered a comparison to other studies slightly challenging. Thus, future studies should try to obtain a balanced dataset. Moreover, future studies should include more hospitals, thus increasing the sample size of the study. Generally, a larger sample size may further improve the performance of the deep learning model.

## 6. Conclusions

This study explored the use of ensemble pre-trained networks, or transfer learning, for the purpose of breast abnormality detection. The model was trained on digital mammograms collected from the department of radiology and of pathology, HUSM. The final ensemble model consisted of a combination of Resnet101, Resnet152, and ResNet50V2. The ensemble model displayed good performance in classifying the suspicious and normal cases across mammographic densities. The provision of this model as a supplementary diagnostic tool to radiologists will reduce their workload. Additionally, the use of this supplementary diagnostic tool in medical workflows will improve the efficiency of breast cancer diagnosis, which, in turn, will accelerate the treatment and management of urgent cases. Furthermore, the use of this model may give radiologists more time to spend on cases classified as suspicious rather than normal. Given the rise in breast cancer incidence, there is a need to improve the efficiency of medical workflows for screening and diagnosing this disease. Thus, the implementation of this model, as a supplementary diagnostic tool for radiologists, in medical workflows will help improve the efficiency of the management and diagnosis of breast cancer.

## Figures and Tables

**Figure 1 diagnostics-13-01780-f001:**
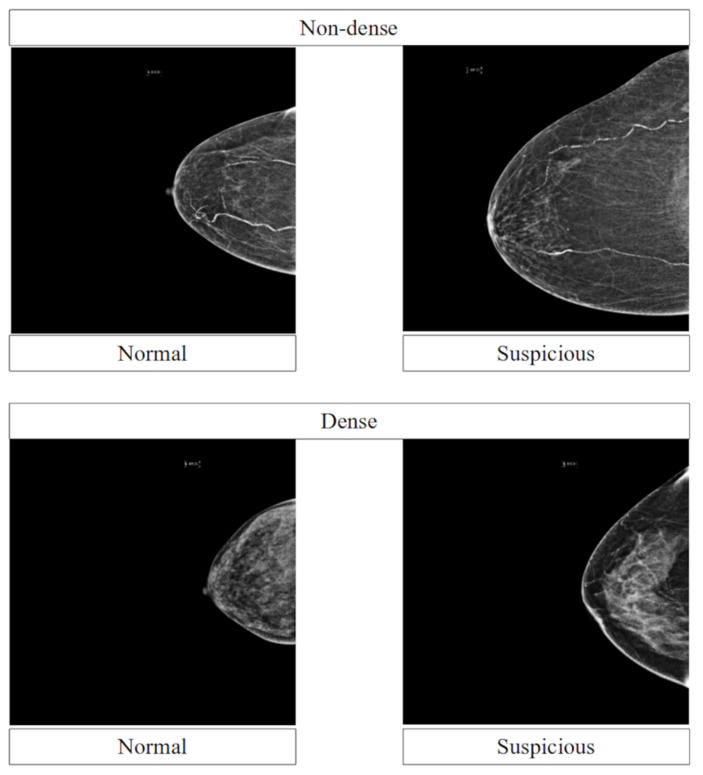
Sample of normal and suspicious mammograms in non-dense and dense groups.

**Figure 2 diagnostics-13-01780-f002:**
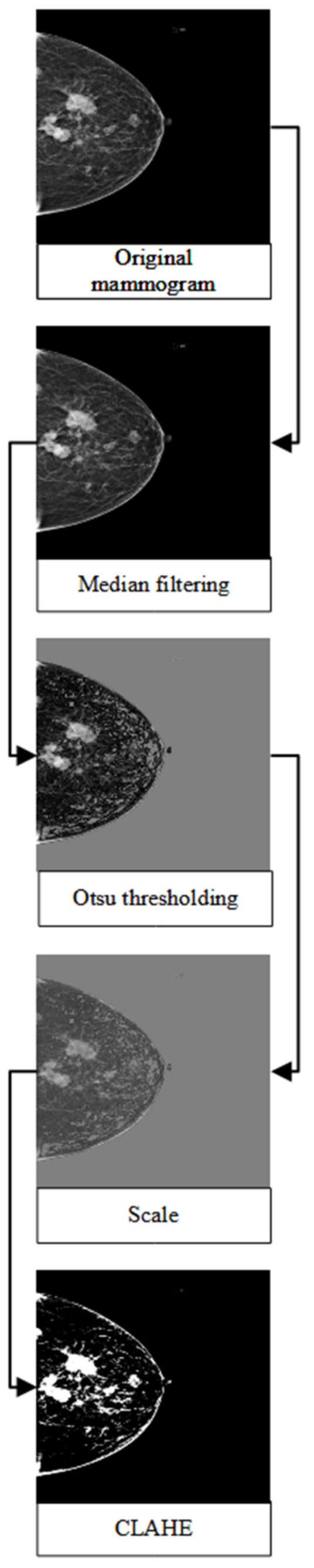
The general flow of the image pre-processing techniques applied to mammograms in this study.

**Figure 3 diagnostics-13-01780-f003:**
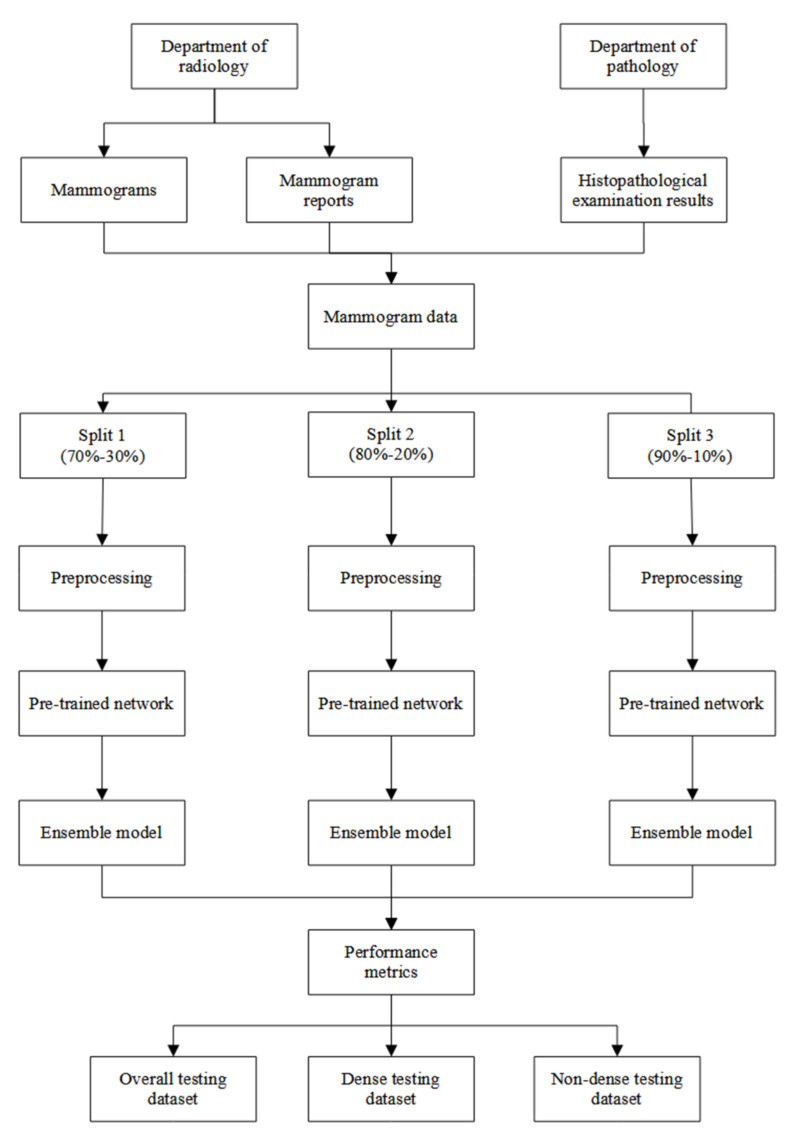
The flow of the analysis in this study.

**Figure 4 diagnostics-13-01780-f004:**
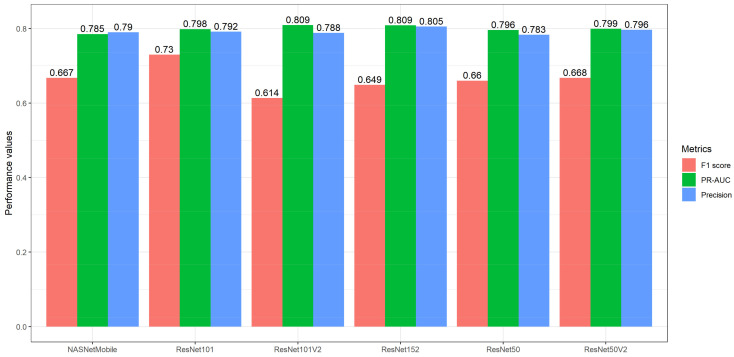
The performance metrics of the top fine-tuned pre-trained networks regarding breast abnormality detection.

**Table 1 diagnostics-13-01780-t001:** Summary of the previous studies related to pre-trained networks and breast cancer classification that utilised digital mammograms.

Study	Database	Pre-Trained Network	Performance Metrics ^1^
Pattanaik (2022) [37]	DDSM	VGG19, MobileNet, Xception, ResNet50V2, InceptionV3, InceptionResNetV2, DenseNet201, DenseNet121, DenseNet121 + ELM ^2^	Accuracy = 0.97 Sensitivity = 0.99 Specificity = 0.99
Khamparia (2021 [27]	DDSM	AlexNet, ResNet50, MobileNet, VGG16, VGG19, MVGG16, MVGG16, ImageNet ^2^	Accuracy = 0.94 AUC = 0.93 Sensitivity = 0.94 Precision = 0.94 F1 score = 0.94
Sabeer (2021) [26]	MIAS	Inception V3, InceptionV2, ResNet, VGG16 ^2^, VGG19, ResNet50	Accuracy = 0.99 AUC = 1.00 Sensitivity = 0.98 Specificity = 0.99 Precision = 0.97 F1 score = 0.98
Ansar (2020) [34]	DDSM CBIS-DDSM	AlexNet, VGG16, VGG19, ResNet50, GoogLeNet, MobileNetV1 ^2^, MobileNetV2	Accuracy = 0.87 Sensitivity = 0.95 Precision = 0.84
Falconi (2020) [30]	CBIS-DDSM	VGG16 ^2^, VGG19, Xception, Resnet101, Resnet152, Resnet50	Accuracy = 0.84 AUC = 0.84 F1 score = 0.85
Falconi (2019) [33]	CBIS-DDSM	MobileNet, ResNet50 ^2^, InceptionV3, NASNet	Accuracy = 0.78
Guan (2019) [28]	DDSM	VGG16 ^2^	Accuracy = 0.92
Mendel (2019) [29]	Primary data	VGG19 ^2^	AUC = 0.81
Yu (2019) [32]	Mini-MIAS	ResNet18 ^2^, ResNet50, ResNet101	Accuracy = 0.96
Mednikov (2018) [36]	INbreast	InceptionV3 ^2^	AUC = 0.91
Jiang (2017) [35]	BCDR-F03	GoogLeNet ^2^, AlexNet	AUC = 0.88
Guan (2017) [31]	MIAS DDSM	VGG16 ^2^	Accuracy = 0.91 AUC = 0.96

^1^ Performance metrics of the best or final model in the study. ^2^ Model with best performance metrics/selected as the final model in the study. DDSM = digital database for screening mammography; MIAS = mammographic image analysis society; CBIS-DDSM = curated breast-imaging subset of database for screening mammography; BCDR-F03 = breast cancer digital repository-film mammography dataset number 3; ELM = extreme learning machine; MVGG16 = modified VGG16.

**Table 2 diagnostics-13-01780-t002:** Performance of fine-tuned, pre-trained networks in terms of detecting breast abnormalities.

Architecture	PR-AUC (Mean, SD)	Precision (Mean, SD)	F1 Score (Mean, SD)	Youden J Index (Mean, SD)
MobileNets	0.79 (0.01)	0.79 (0.00)	0.49 (0.07)	0.02 (0.01)
MobileNetV2	0.79 (0.00)	0.79 (0.01)	0.46 (0.11)	0.02 (0.04)
MobileNetV3Small	0.80 (0.01)	0.81 (0.02)	0.56 (0.09)	0.06 (0.04)
NASNetLarge	0.80 (0.03)	0.80 (0.03)	0.68 (0.09)	0.06 (0.09)
NASNetMobile	0.79 (0.02)	0.79 (0.02)	0.67 (0.06)	0.03 (0.05)
ResNet101	0.80 (0.03)	0.79 (0.01)	0.73 (0.08)	0.04 (0.04)
ResNet101V2	0.81 (0.01)	0.79 (0.01)	0.61 (0.07)	0.02 (0.03)
ResNet152	0.81 (0.01)	0.81 (0.01)	0.65 (0.04)	0.07 (0.03)
ResNet152V2	0.80 (0.03)	0.80 (0.03)	0.60 (0.17)	0.07 (0.07)
ResNet50	0.80 (0.03)	0.78 (0.02)	0.66 (0.08)	0.01 (0.03)
ResNet50V2	0.80 (0.03)	0.80 (0.01)	0.67 (0.01)	0.05 (0.03)
VGG16	0.79 (0.03)	0.77 (0.04)	0.61 (0.14)	−0.01 (0.08)
VGG19	0.78 (0.02)	0.78 (0.01)	0.57 (0.11)	0.00 (0.04)

PR-AUC = precision–recall area under the curve. SD = standard deviation.

**Table 3 diagnostics-13-01780-t003:** Performance comparison between the ensemble transfer learning model and the individual models with respect to detection of breast abnormalities.

Model	Precision (Mean, SD)	F1 Score (Mean, SD)	Youden J Index (Mean, SD)
Ensemble model 1	0.81 (0.01)	0.65 (0.01)	0.09 (0.03)
Ensemble model 2	0.81 (0.01)	0.66 (0.01)	0.09 (0.04)
Ensemble model 3	0.82 (0.01)	0.68 (0.01)	0.12 (0.03)
NASNetMobile	0.79 (0.02)	0.67 (0.06)	0.03 (0.05)
ResNet101	0.79 (0.01)	0.73 (0.08)	0.04 (0.04)
ResNet101V2	0.79 (0.01)	0.61 (0.07)	0.02 (0.03)
ResNet152	0.81 (0.01)	0.65 (0.04)	0.07 (0.03)
ResNet50V2	0.80 (0.01)	0.67 (0.01)	0.05 (0.03)

PR-AUC = precision–recall area under the curve. SD = standard deviation. Ensemble model 1 = Resnet101 + NASNetMobile + ResNet50V2. Ensemble model 2 = Resnet101V2 + Resnet152 + ResNet50V2. Ensemble model 3 = Resnet101 + Resnet152 + ResNet50V2.

**Table 4 diagnostics-13-01780-t004:** The descriptive performance of the final ensemble model across breast densities on the overall, dense, and non-dense testing datasets.

Metrics	Overall	Dense	Non-Dense
Precision	0.82 (0.01)	0.86 (0.01)	0.77 (0.00)
F1 score	0.68 (0.01)	0.75 (0.01)	0.60 (0.02)
Youden J Index	0.12 (0.03)	0.21 (0.04)	0.03 (0.03)
Sensitivity	0.58 (0.02)	0.67 (0.01)	0.49 (0.03)
Specificity	0.54 (0.02)	0.54 (0.03)	0.54 (0.01)

**Table 5 diagnostics-13-01780-t005:** The performance comparison of the final ensemble model between dense and non-dense breast testing datasets using Wilcoxon rank sum statistical test.

Metrics	Dense Median (IQR)	Non-Dense Median (IQR)	W Statistics	*p* Value
Precision	0.86 (0.01)	0.77 (0.00)	9	0.1
F1 score	0.75 (0.01)	0.60 (0.02)	9	0.1
Youden J Index	0.22 (0.04)	0.03 (0.03)	9	0.1
Sensitivity	0.67 (0.01)	0.49 (0.03)	9	0.1
Specificity	0.55 (0.03)	0.54 (0.01)	6	0.7

IQR = interquartile range.

## Data Availability

The data are available upon reasonable request to the corresponding author.

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
