# Peer review of "Developing a Supplementary Diagnostic Tool for Breast Cancer Risk Estimation Using Ensemble Transfer Learning"

_diagnostics, 2023, doi:10.3390/diagnostics13101780_

Round 1

Reviewer 1 Report

Peer Review Report

Manuscript ID: Diagnostics-2282674

Title: Developing a supplementary diagnostic tool for breast cancer risk estimation using ensemble transfer learning

The Manuscript by Hanis et al. namely “Developing a supplementary diagnostic tool for breast cancer risk estimation using ensemble transfer learning” lies within the journal’s scope of diagnostics. The authors present a diagnostic tool for breast cancer risk estimation using artificial intelligence. The study is of medical significance as it can help physicians to detect the abnormality of breast in earlier stages. The study mainly focus on Mammography which is a two-dimensional image data sets. The study employs thirteen pre-trained networks including only six of them are reported in abstract namely Net101V2, ResNet152, Mobile NetV35Small, ResNet152, ResNet101 etc. The study didn’t explain other used networks in detail. What is the difference between all of these pre-trained networks? The authors needs to clear whether the pre-trained networks can only differentiates breast densities as dense and non-dense or they can further classify and differentiates them per BIRAD definition. Breast Imaging-Reporting and Data System (BI-RADS) classifies breasts into four categories in the order of radiographic breast density composition namely extremely dense (ED)-10% (type-D), heterogeneously dense (HD)-40"/o (type-C), scattered fibroglandular (SF)-40% (type-B) and predominantly fatty (PF) is reported in about 10% of women (type-A) 1 • Under such classification BI-RADS defines tumour into namely six assessment categories2 • TNM (Tumour-NodeMetastasis) tool is widely used to describe the stage of tumour. T refers to size of the tumour, N relates to primary inspection whether cancerous cells spread to lymphatic nodes and M explains whether it invades to other organs of the patient. Clinically, the staging of the tumour in the breast may be defined as Tl (size:s20 mm): Tlmi ('.S 1 mm), Tla (1 mm < size '.S 5 mm), Tlb (5 mm< size '.S 10 mm), Tlc (10 mm< size '.S 20 mm); T2 (20 mm < size '.S 50 mm); T3 (2': 50 mm); T4a (tumour has grown into chest wall), T4b (tumour has grown into skin), T4c (grown into chest wall and skin), T4d (inflammatory). Refer [https://doi.org/10.31224/osf.io/enx4r]

1) Introduction and Literature Review:

1a) The introduction is very short. the authors should extend the abstract's length.

1b) The authors should merge Literature Review with Introduction and modify the structure of manuscript accordingly.

================================

2) Keywords and Highlights:

2a) The authors must edit old keywords in the article. Consider adding additional keywords to enhance the scope of search of Manuscript.

2b) The authors must add Highlights in the article.

================================

3) Abstract:

3a) The abstract doesn’t have novelty in it. The authors should rewrite the abstract with main novelty in it.

3b) What is the main purpose of the article? The authors should focus on novelty on this section. Please highlight it.

================================ 

4) Results

Add caption details for figure 03. The y-axis name should be modified.

5.) Discussions:

5a) It has some figures, but technical description for figures is not enough. The authors must describe as well for every figure (major comment).

5b) this section need to update and need more detail.

5c) The authors should add more detail and use more Graphs in this sections.

================================

6) Conclusion and recommendations:

6a) Conclusion lack of novelty. Please rewrite your conclusion and add some highlight and novelty in it (major comment).

6b) Conclusion is so short the authors should extend the material.

================================

7) Graphical abstract:

7a) In my opinion, the graphical abstract can help researchers. I suggest authors add one Graphical abstract in next version

================================

8) References:

8a) References should be updates (2021-2022)

9) The authors are advised to present a hierarchy under which the pre-trained networks were developed and mention its functioning.

10) Also discuss briefly if these pre-trained networks can be able to differentiate Magnetic Resonance Images. There are other commercial software available to construct the three-dimensional anatomy of tumor and breast itself. All feature information in terms of depth and volume can be visualized in three-dimensional view by surgeons. Discuss how artificial intelligence governed pre-trained networks can use information from grayscale mammogram images can be extended  to MRI images. Discuss in reference to [https://doi.org/10.1016/j.cmpb.2020.105781] in the discussion section.

Reviewer 2 Report

1. Introduction needs to be improved. The novelty and the contribution of the authors are not clearly mentioned.

2. The paper flow needs to be added in the introduction.

3. Sample images of the original dataset can be added.

4. The authors can add a justification for using these pre-trained models for testing purposes.

5. The confusion matrix needs to be added.

6. Accuracy curve, loss curve, and roc curve along with AUC values need to be shown in the paper.

7. The results and the discussion section need to be more elaborate and convincing. There is a lack of experimental results and an explanation of the results is also missing.

8. Table 5 metrics need to be explained for a better understanding of the readers.

9. A valid justification on choosing the ensemble models in the experiments is missing. 
